# The Activity of N-acetyl-β-hexosaminidase in the Blood, Urine, Cerebrospinal Fluid and Vitreous Humor Died People Due to Alcohol Intoxication

**DOI:** 10.3390/jcm9113636

**Published:** 2020-11-12

**Authors:** Iwona Ptaszyńska-Sarosiek, Sylwia Chojnowska, Sławomir Dariusz Szajda, Michał Szeremeta, Zofia Wardaszka, Urszula Cwalina, Anna Niemcunowicz-Janica, Napoleon Waszkiewicz

**Affiliations:** 1Department of Forensic Medicine, Medical University of Bialystok, Waszyngtona 13 Str., 15-269 Białystok, Poland; i.sarosiek@wp.pl (I.P.-S.); z.wardaszka@gmail.com (Z.W.); annjanica@gmail.com (A.N.-J.); 2Medical Institute, College of Computer Science and Business Administration, Akademicka 14 Str., 18-400 Łomża, Poland; sylwiacho3@gmail.com; 3Department of Psychiatry, Medical University of Bialystok, Plac Brodowicza 1 Str., 16-070 Choroszcz, Poland; sbszajda@gmail.com (S.D.S.); napoleonwas@yahoo.com (N.W.); 4Department of Statistics and Medical Informatics, Medical University of Bialystok, Szpitalna 37 Str., 15-295 Białystok, Poland; urszula.cwalina@umb.edu.pl

**Keywords:** hexosaminidase, ethanol intoxication, blood, vitreous humor, cerebrospinal fluid, urine

## Abstract

Background: The article aimed to assess the activity of the hexosaminidase (HEX) and its HEX A and HEX B isoenzymes in persons who suddenly died due to ethanol poisoning and explain the cause of their death. Methods: The research involved two groups of the deceased group A—22 people (20 males, 2 females; the average age 46 years) who died due to alcohol intoxication (with the blood alcohol content of 4‰ and above in all biological materials at the time of death—blood, urine, cerebrospinal fluid, and vitreous humor), and group B—30 people (22 males, 8 females; the average age 54 years), who died suddenly due to other reasons than alcohol. Results: The highest activity of the HEX was found in the serum of A and B groups. A significantly lower activity of HEX, HEX A, and HEX B was observed in the urine of group A in comparison to the sober decedents. Conclusion: The lower activity of HEX and its isoenzymes in the dead’s urine due to ethanol poisoning may suggest its usefulness as a potential marker of harmful alcohol drinking. Damage done to the kidneys by ethanol poisoning may be one of the possible mechanisms leading to death. Kidneys may be damaged intravitally via the inflammatory agent. Thus, it is necessary to conduct further research to evaluate the diagnostic usefulness of exoglycosidases while determining the death mechanisms of people who lost their lives due to ethanol poisoning.

## 1. Introduction

Harmful alcohol consumption (abuse) is a serious and growing social and economic health problem worldwide. It is estimated that 3.8% of all deaths worldwide and 4.6% of the global burden of disease and trauma can be attributed to alcohol consumption leading to psychiatric disorders, cancers, cardiovascular diseases, or cirrhosis [1]. Additionally, an alcoholic hangover, which can happen after an episode of heavy drinking when blood ethanol concentration (BAC) returns to 0.0% [2], can negatively affect people’s mental and physical well-being, increasing the number of accidents and injuries [3]. Up to 30% of all hospital admissions and health-care costs may be attributable to alcohol abuse [4,5]. Even at the alcohol-dependence stage, physicians are likely to identify only 20–50% of patients attending medical care [6]. About 52% of the world’s adult population consumes alcohol; alcohol dependence affects 2–4% of people globally, while heavy drinking (binge drinking) is reported by about 16–18% of drinkers. Worldwide, the highest alcohol consumption levels per capita were found within the European region [7,8,9]. Binge drinkers occasionally drink more than 5 units of alcohol/day (single occasion drinking). An increased likelihood of binge drinking is seen in stressed individuals, who have also presented changes in the activity in the hypothalamic–pituitary–adrenal (HPA) axis [10]. When binge drinking results in harm, it is then called harmful drinking or alcohol abuse. Alcohol-dependent persons chronically drink alcohol with physical and/or psychological dependence. Therefore, early screening of binge drinking is especially important. It is a much more common problem than chronic drinking and may precede the sequence of events leading to alcohol addiction [11,12]. 

A recognized indicator of chronic alcohol use is lysosomal exoglycosidase N-acetyl-β-hexosaminidase (HEX) (E.C.3.2.1.52) [13,14], abstracting N-Acetylglucosamine and N-acetylgalactosamine from a non-reducing end of oligosaccharide: glycolipids, glycoproteins, and proteoglycans [15,16]. 

The presence of HEX was found in the kidneys, spleen, liver [17,18], gastric mucosa, the mucous membrane of the intestine, cerebral cortex, skin fibroblasts, placenta, and lungs [19,20], in cancerous tissues, including in cancer tissues of the liver, pancreas [21], and a large intestine [22]. HEX was also determined in body fluids: serum, urine [23,24,25], in cerebrospinal fluid, in synovial fluid [26,27,28,29], in the saliva [30], as well as in many other tissues and human organs [31,32]. 

Determining the activity of N-acetyl-β-hexosaminidase in the serum [33] and the urine [34] was applied in the diagnostics of alcohol dependence. It was stated that the action of HEX B isoenzyme in the serum, total HEX in the urine, and HEX A in the saliva are a sensitive marker of chronic ethanol consumption [6,35]. HEX activity increases in the urine and blood of people dependent on alcohol and persons after the consumption of the amount higher than 60 g a day for at least 10 days [6] and returns to the normal state after 7–10 days of sobriety in the serum and urine, and after 4 weeks in the saliva [12,36]. Although HEX has limited usefulness in a total population, its activity in the blood serum, saliva, and urine may increase after single occasional drinking (binge drinking) with a high amount (2 g/kg) of ethanol [37]. 

Acute ethanol intoxication (binge drinking) negatively influences numerous organs, including the brain, salivary glands, the heart, the liver, and kidneys [38,39,40,41,42,43]. Chronic alcohol drinking leads to some changes in the mental state and behavior and functional and morphological disorders in the central nervous system, the cardiovascular system, the immunological system, the liver, pancreas, and endocrine organs [44,45]. 

The studies concerning the HEX activity in many biological materials were conducted only in live people [46,47,48,49,50]. There is a lack of research concerning HEX activity in any biological material derived from people who died due to alcohol intoxication. There is also little evidence concerning the damage to kidneys due to ethanol poisoning, especially the mechanisms in which the damage may occur [51]. Therefore, this study aims to evaluate the HEX and its isoenzymes HEX A and HEX B in people who died of ethanol intoxication and an attempt to determine the mechanism of their death. 

## 2. Material and Methods

### 2.1. Subjects

The research was conducted on 2 groups of the deceased: the first (A) consisted of 22 people (20 males and 2 females; range 26–82 years old, the average age 46 years) who died due to ethanol intoxication; the second (B) included 30 people (22 males and 8 females; range 15–83 years old, the average age 54 years), in whose bodies no alcohol was found. All individuals in group B died as a result of suicide, traffic accident or other unfortunate events. In all cases, the presence of illnesses of kidneys, livers, cancers, and chronic inflammatory diseases—including rheumatoid arthritis—were ruled out. The family interview ruled out that poisoned people would consume alcohol before their last consumption. 

### 2.2. Ethics

The research was conducted with the Local Ethical Committee of Medical University of Bialystok (act no.: R-I-002/82/2013).

### 2.3. Procedures

The material was collected during the autopsy with a syringe, in the amount of 5 mL each from the following:the femoral vein (blood)the bulb of the eye (vitreous humor)the lateral ventricle of the brain (cerebrospinal fluid)the urinary bladder (urine)

The concentration of the ethanol was determined by the gas chromatography method, headspace technique (HS-GC-FID). After the samples were derived to determine the ethanol concentration, the rest of the biological material was centrifuged at 3000× *g* for 20 min at 4 °C. The supernatant was divided, frozen, and stored in Eppendorf tubes at −80 °C until the biological analysis was performed. A Thermo Electron Corporation Trace, GC Ultra chromatograph, is equipped with an FID detector and headspace TriPlus automatic injector, with a capillary column a ZB-BAC1 (30 m × 0.32 mm ID × 1.8 μm film thickness) and a ZB-BAC2 (30 m × 0.32 mm ID × 1.2 μm film thickness). The following conditions were used: carrier gas—helium 1.8 mL/min., column temperature—40 °C, injector temperature—150 °C, detector temperature—200 °C, sample heating temperature—60 °C, thermoregulation time—5 min. Using this method allows detecting the metabolite of alcohol: acetaldehyde.

### 2.4. Assays

Activities (pKat/mL) of N-acetyl-β-d-hexosaminidase and its A and B isoenzymes of liquids were determined by the colorimetric method of Chojnowska et al. [52], based on the determination of p-nitrophenol released from p-nitrophenyl-derivatives of the respective sugars. As a substrate for determining the HEX, 4-nitrophenyl-β-d-N-acetylglucosaminopyranoside (Sigma, St. Louis, MO, USA) was used. Measurements of p-nitrophenol released by the respective exoglycosidases were carried out at 405 nm using the microplate reader Elx800TM. The activities (pKat/mL) were assayed in duplicate, and the means were used as final values. Heat stable HEX B isoenzyme of HEX was measured after selective heat denaturation of the thermolabile HEX A. HEX A was calculated from the difference between the total HEX and HEX B activity. 

### 2.5. Statistics

Statistical analysis was performed with Statistica version 12.5 (StatSoft, Cracov, Poland). All data were tested for normal distribution. Descriptive characteristics are presented as means ± SD. Nonparametric results were expressed as median (IQR or minimum–maximum). The differences between control (B) and alcohol-intoxicated (A) group were evaluated using the Mann–Whitney “U” test. Comparisons between groups were made using Kruskal–Wallis analysis with posthoc tests. Spearman’s rank correlation coefficient was used to measure the statistical dependence between nonparametric variables. Statistical significance was defined as *p* < 0.05.

### 2.6. Results

In the B group of the deceased, no ethyl alcohol was found in all the samples. In the research group (A) the concentration of ethanol in the serum ranged from 4‰ to 4.5‰, in the vitreous humor—from 4‰ to 4.9‰, in the cerebrospinal fluid—from 4‰ to 5.3‰ and in the urine—from 4‰ to 6.1‰. In all the studied fluids (the serum, cerebrospinal fluid, vitreous humor, and urine), the ethanol concentration was observed to be higher than 4‰. 

Because the number of women analyzed was low in the A group, and no differences in HEX and its isoenzymes were found in the B group, sex of the dead was not considered during the primary statistical analyses. The comparison of the control group (B) and the study group (A) on the ground of age and the activity of HEX, HEX A, and HEX B in particular studied materials was performed (Table 1, Figure 1). Group A and B did not differ significantly about age (Table 1, Figure 1). A significantly lower activity of HEX (*p* = 0.015), HEX A (*p* = 0.011), and HEX B (*p* = 0.015) were found in the urine of group A in comparison to group B. No significant statistical differences were observed in other materials. 

It was observed that the blood serum had a significantly higher activity of HEX, HEX A, and HEX B concerning all the studied materials (vitreous humor, cerebrospinal fluid, and urine) in both groups A and B (Figure 1, Table 1). The authors in Figure 1 presented a comparison of the enzymatic activity between groups A and B separately in all types of materials. Only the urine was the fluid that showed statistically significant differences between the control group and the study group for each of the tested enzymes.

In the B group, a positive correlation was observed between HEX B in the cerebrospinal fluid and HEX (r = 0.495, *p* = 0.010), HEX B (0.403, *p* = 0.041) and HEX A (r = 0.500, *p* = 0.009) in the serum as well as between HEX in the serum and HEX (r = 0.503, *p* = 0.005) and HEX B (r = 0.426, *p* = 0.019) in the vitreous humor. Moreover, a correlation was observed between HEX A in the serum and HEX B (r = 0.478, *p* = 0.007) and HEX A (r = 0.404, *p* = 0.027) in the vitreous humor. A correlation was also found between HEX A in the blood serum and HEX in the vitreous humor (r = 0.589, *p* < 0.001). In addition, it was also found that there is a correlation between HEX and HEX B in the blood serum (r = 0.542, *p* < 0.001), a high correlation between HEX and HEX A in the blood serum (r = 0.975, *p* < 0.001) and a correlation between HEX B and HEX A in the blood serum (r = 0.415, *p* < 0.05). In the A group, a correlation was observed between HEX B in the blood serum and HEX (r = 0.654, *p* < 0.001) and HEX A (r = 0.651, *p* = 0.001) in the cerebrospinal fluid and a high correlation between HEX and HEX A in the serum (r = 0.938, *p* < 0.001). A negative correlation was found between HEX B in the serum and HEX in the vitreous humor (r = −0.460, *p* = 0.031) (Table 2). The blood ethanol concentration (BAC) significantly correlated with the activity of HEX B in the serum (r = 0.495, *p* = 0.019). A positive statistically significant correlation with the strength r = 0.495 was found between the concentration of alcohol and HEX B in the blood (Table 3).

## 3. Discussion

It is known that alcohol dependence syndrome includes alcohol tolerance changes, which is exceptionally high during the first years of dependence [53,54]. Alcohol dependence is a severe mental illness, and there is a need for more effective preventive and therapeutic strategies [55]. 

The highest concentration of ethanol in the urine of group A, compared to other biological materials (serum and vitreous humor in particular), may indicate a selective accumulation of ethanol in the urine during intoxication. On the other hand, it may be a sign of the most significant kidneys damage, more severe than the brain’s damage or the eye. It is known that the organ damage caused by the ethanol is due to the ethanol action itself or due to its metabolites [16]. It also seems relevant that ethanol retention in the bladder enables us to detect ethanol in urine even a few hours longer than in the blood [56,57]. 

It was observed that HEX, and B isoenzyme in the blood serum, the total HEX in the urine, and HEX A in the saliva, are very sensitive markers of chronic alcohol consumption [13,35,36,58]. The reasons for the increase in the activity of HEX and its isoenzymes in bodily fluids after ethanol consumption may be an increased permeability of the lysosomal membrane and the escape of the exoglycosides from the lysosomes, and then from the cells to the bodily fluids, a delayed clearing of HEX and its isoenzymes from the bodily fluids, disordered transport of the enzyme to organelles, the increased synthesis of HEX and its isoenzymes by activated leucocytes or leakage from the damaged cells [15,58,59]. We also proved that HEX and its isoenzymes A and B were higher only in the deceased’s blood serum in relation to the vitreous humor, cerebrospinal fluid, and urine (Table 1, Figure 1). The increase in the activity of HEX in the serum may be connected with the liver damage, which had been indicated earlier, as well as with a lower clearing of lysosomal enzymes from the blood [35,58]. In living people, HEX A activity significantly increases in urine even after a single drinking episode [37,49]. In people who died due to ethanol intoxication, we observed a lower activity of HEX, HEX A, and HEX B in the urine, cerebrospinal fluid, and vitreous humor in comparison to the blood serum (Table 1, Figure 1), which may be the proof of damage to the blood–organ barrier. Other authors explained that the growth of HEX activity in urine was caused by permeation from the blood, and its high activity resulted from liver damage caused by the alcohol [6,13]. In our research, the decrease in HEX in urine could be explained by harmful alcohol use, not alcohol dependence. An earlier study [60], indicated that occasional “binge drinking” leads to brain damage faster than chronic drinking. Acute ethanol intoxication causes the impairment of the functions of kidneys and the clearing of creatinine. Even the next day after alcohol intoxication, a transient dysfunction of kidneys may occur [39,59]. In the deceased intoxicated with alcohol, significantly lower levels of HEX, HEX A, and HEX B observed in the urine than the blood serum, may indicate significant damage to the kidneys caused by ethanol and its metabolites after their lower secretion to the urine. Isoenzyme HEX A was described earlier as a specific organ damage indicator through the inflammatory process [58]. Our study showed the highest HEX A activity in the urine of people who died due to ethanol intoxication compared to the HEX and HEX B, which may damage the kidneys caused by the ethanol-induced inflammatory mechanism. An earlier study Waszkiewicz et al. [37] showed that at 108 h after binge drinking, there was a significant (by the app. 1/3) growth of HEX, HEX A, and HEX B activity in the serum and a significant increase in only HEX A in the urine. There was also a reversed correlation between HEX B’s concentration in the blood serum and urine after binge session. 

The organ that is particularly exposed to ethanol’s toxic influence and its metabolites is the liver, whose function is to deal with metabolic changes of alcohol [61,62]. Liver damage caused by ethanol’s chronic consumption is also noticeable in laboratory tests [63,64,65,66,67]. In alcoholic-induced hepatitis, increased activity of gamma-glutamyltra/nsferase (GGT), aspartate transaminase (AST), alanine transaminase (ALT), etc. was observed in the blood [45]. Post-alcohol liver impairment may disturb kidneys’ function by fluctuations in plasma and glomerular filtration [68]. Nicholson and Taylor [69] researched dogs, which showed that a single dose of ethanol (3 g/kg of body weight) increased the plasma volume from 10 to 26 h after ethanol consumption. The decrease in the activity of HEX and its isoenzymes in the urine confirmed by our research may also be explained by a lower secretion of vasopressin (antidiuretic hormone) in people consuming big amounts of ethanol, leading to urine dilution [41]. On the other hand, the attenuation of the magnesium concentration (enzyme cofactor) in chronic drinkers [41] leads to a decrease in the enzyme activity, which to some degree, might be the reason for the reduction of HEX activity in the urine in our study. 

During chronic alcohol intoxication, a more significant increase in the activity of HEX B was discovered in the serum [49,70]. In contrast, heavy drinkers and nondrinkers had a higher level of HEX A than HEX B even after moderate consumption of alcohol [49]. Similarly, our research showed that people who died due to ethanol intoxication had serum activity of HEX A significantly higher than HEX B (Table 1). It might be the result of a short-time and a high-dose alcohol-induced supravital increase in the serum activity of thermolabile isoenzyme HEX A [69,71] and its related inflammatory organ impairment. 

Although in the time of death, there is an increase in the activity of the lysosomal enzymes in the body fluids, the decreased activity of HEX, HEX A, and HEX B in the urine of the alcohol-intoxicated individuals may be due to the potential inactivation of HEX and its isoenzymes in the urine by the ethanol and/or its metabolites, acetaldehyde in particular [72]. It may be confirmed that a significantly higher concentration of ethanol was discovered in the urine. In physiological conditions (control group), enzymes can freely penetrate the blood–eye barrier (HEX, HEX A) and the blood–brain barrier (HEX B), which we showed on the basis of correlations of these isoenzymes in serum and body fluids. We found no correlation in the alcohol-intoxicated group in HEX, HEX A, nor HEX B isoenzyme between-fluids. Still, we found a statistically significant correlation between serum HEX B activity and ethanol concentration in the blood (r = 0.495, *p* = 0.019). Hence, we can conclude that these barriers may become impaired (less porous). Lack of correlation between ethanol concentration and the lysosomal exoglycosidases activity in other body fluids (except serum), maybe since ethanol itself causes the most extensive damage to the liver, and from this organ, exoglycosidases may be released to the blood. The impairment of other tissues (brain, vitreous humor, kidneys) may be caused, preferably by ethanol metabolites. 

Modern medicine does not offer a comprehensive diagnosis of fatal alcohol intoxication based on exoglycosidases changes in various tissues or body fluids. The hope for such can be our study containing an analysis of many exoglycosidases and their behavior in four body fluids resulting from alcohol intoxication in the deceased. However, it should be recalled that biomarker research may also be useful in psychiatric disorders [73]. 

Our present study also has some limitations: (a) small study groups (b) no other biomarkers (except for ethanol) have been included in the study, which might give us more useful information about intoxication. Therefore, some of the parameters in our study’s decreased values might also be due to the disturbed metabolism of enzymes and/or the rapid catabolism/elimination, which we did not investigate. This may be another limitation of our study. 

## 4. Conclusions

1. The impairment of the kidneys during ethanol intoxication may be one of the most likely death mechanisms, and kidneys might be damaged supravitally due to inflammation.

2. The decrease in the HEX activity and its isoenzymes in the urine of people who died due to ethanol intoxication may be a potential marker of harmful alcohol use. 

3. It is a requisite to conduct further research on the diagnostic usefulness of exoglycosidases to determine death mechanisms in people intoxicated with ethanol. 

## Figures and Tables

**Figure 1 jcm-09-03636-f001:**
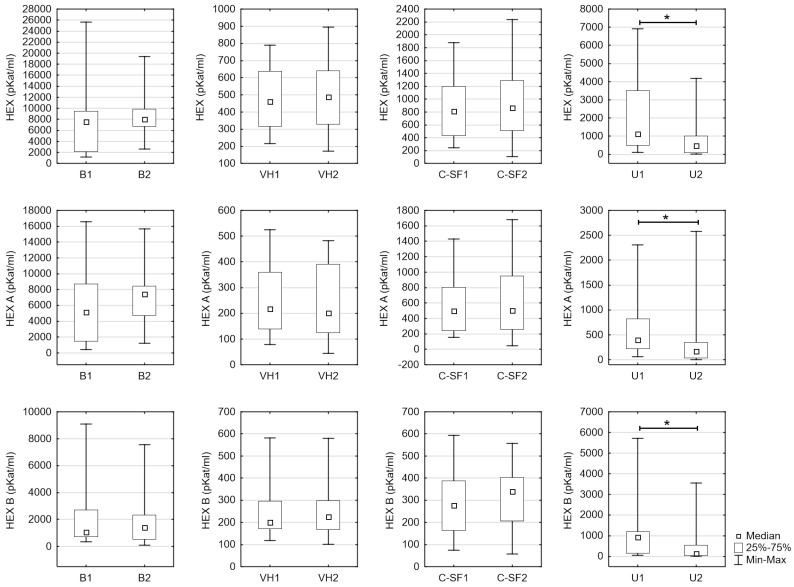
Comparison of the enzymatic activity between both groups separately in all types of fluids (no. “1”—sober individuals, no. “2”—group with a fatal ethanol intoxication).

**Table 1 jcm-09-03636-t001:** Activity of hexosaminidase (HEX), HEX A, and HEX B concerning all studied materials.

Variable	Material	B	A	*p*
Age		54 (15–83)	46 (26–82)	0.416
HEX [pKat/mL]	Blood serum	7535.58 (1162.25–25,679.50)	8035.68 (2638.80–19,378.00)	0.198
	Vitreous humor	460.57 (215.26–788.23)	488.11 (170.57–895.26)	0.919
	Cerebrospinal fluid	813.72 (246.31–1878.45)	488.11 (104.93–2235.65)	0.756
	Urine	1124.60 (106.23–6923.41)	472.51 (13.86–4179.39)	0.015 *
HEX A [pKat/mL]	Blood serum	5132.45 (439.20–16,588.20)	7436.98 (1221.60–15,665.60)	0.162
	Vitreous humor	217.00 (78.43–524.30)	200.92 (44.79–482.48)	0.788
	Cerebrospinal fluid	501.02 (152.11–1429.15)	501.10 (48.10–1678.73)	0.934
	Urine	394.89 (60.54–2311.70)	175.56 (2.99–2581.15)	0.011 *
HEX B [pKat/mL]	Blood serum	1071.46 (364.22–9091.30)	1408.80 (102.91–7562.05)	0.875
	Vitreous humour	201.11 (118.23–581.93)	225.26 (100.64–580.41)	0.399
	Cerebrospinal fluid	277.69 (74.16–594.25)	339.97 (56.84–556.92)	0.321
	Urine	937.56 (45.69–5711.90)	133.85 (7.77–3542.55)	0.015 *

* *p* < 0.05.

**Table 2 jcm-09-03636-t002:** Correlation between HEX, HEX A, and HEX B concerning all studied materials.

		B Group Blood Serum				A Group Blood Serum			
Variable	Mate-Rial	HEX	HEX A	HEX B	Variable	HEX	HEX A	HEX B	Blood Ethanol Concentration (BAC)
HEX	Blood serum	-	0.975 ***	0.542 **	HEX	-	0.938 ***	0.345	0.407
	Vitreous humor	0.503 **	0.589 ***	−0.160		0.285	0.299	−0.460 *	0.031
	Cerebro-spinal fluid	0.204	0.174	0.357		0.103	−0.061	0.654 ***	0.316
	Urine	0.229	0.283	0.138		−0.010	0.073	−0.166	−0.204
HEX A	Blood serum	0.975 ***	-	0.415 *	HEX A	0.938 ***	-	0.186	0.293
	Vitreous humor	0.318	0.404 *	−0.237		0.046	0.028	−0.404	−0.003
	Cerebro-spinal fluid	0.066	0.044	0.247		−0.015	−0.178	0.651 **	0.297
	Urine	0.151	0.230	−0.056		0.055	0.082	−0.108	−0.040
HEX B	Blood serum	0.542 **	0.415 *	-	HEX B	0.345	0.184	-	0.495 *
	Vitreous humor	0.426 *	0.478 **	−0.073		0.415	0.392	−0.286	0.130
	Cerebro-spinal fluid	0.495 *	0.500 **	0.403 *		0.300	0.179	0.391	0.367
	Urine	0.200	0.247	0.199		−0.031	0.061	−0.149	−0.233

* *p* < 0.05; ** *p* < 0.01; *** *p* < 0.001.

**Table 3 jcm-09-03636-t003:** Statistically correlation between alcohol concentration and activity of HEX, HEX A, and HEX B in all studied materials.

	HEX	HEX A	HEX B
Blood serum	0.407	0.293	0.495 *
Vitreous humor	0.052	−0.186	0.176
Cerebrospinal fluid	0.197	0.230	0.125
Urine	0.014	0.099	−0.046

* *p* < 0.05.

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
