# Peer review of "The Activity of N-acetyl-β-hexosaminidase in the Blood, Urine, Cerebrospinal Fluid and Vitreous Humor Died People Due to Alcohol Intoxication"

_jcm, 2020, doi:10.3390/jcm9113636_

Round 1

Reviewer 1 Report

  1. The authors performed analysis of the activity of the hexosaminidase (HEX) and its HEX A and HEX B isoenzymes in persons who suddenly died due to ethanol poisoning and the results of the analysis of the results showed that the lower activity of HEX and its isoenzymes in the dead's urine due to ethanol poisoning may suggest its usefulness as a potential marker of harmful alcohol drinking.
  2. The study addresses a clear knowledge gap, investigating the markers of harmful alcohol drinking.
  3. However, I would like to notice some minor questions, in order to improve the quality of the paper. 

a In Introduction section (line 73): the authors use word “psyche” I would recommend substitution with “mental state”

  1. In Material and methods section (line 87) two groups are described A and C and the reader is concerned about the group B – it would be more clear if the groups are A and B  
  2. The limitation of the study is a small number of women – it would be very interesting to have results in the group of women in future research.

Author Response

Dear Editor

Responding to critical comments, we kindly explain the following:

  1. Term "psyche" was changed to the term "mental state" (line 73)
  2. The study's limitations result from the actual conditions of alcohol
    consumption in the social structure in which the overwhelming majority
    are men.
  3. Throughout the manuscript, group C has been changed to B.

Yours sincerely
Corresponding author.

Reviewer 2 Report

Comments

In the current manuscript authors attempted to assess the activity of the hexosaminidase (HEX) and its HEX A and HEX B isoenzymes in many biological materials in persons who died due to ethanol intoxication. Some comments related to manuscript are highlighted below:

  1. BCA will be sober if a person had not consumed alcohol in last few days. Therefore, I would like to ask authors the previous history of deceased (both groups A and C) of alcohol consumption.
  2. In the study, author had included a wide range of age group (15-83). Thus, there would be the variability in the data. Author must justify why they choose wide range of age in the study?
  3. Authors measured the alcohol content in the body fluids using technique of head space analysis. I would like to ask author to briefly describe the protocol here. I also would like to know whether using this technique, is it possible to detect the metabolites of alcohol as well.
  4. Further, In Table 2, authors did correlation study of enzyme HEX and its isoforms HEX A and HEX B between fluids (the serum, cerebrospinal fluid, vitreous humor, and urine) of group A and C. Here, I would suggest author to perform correlation study between alcohol conc and enzyme HEX and isoforms HEX A and HEX B in various type of fluids studied here.
  5. In figure 1, I would suggest authors to compare the enzymatic activity between the group A and C separately in all type of fluids.

Author Response

Dear Editor

Responding to critical comments, we kindly explain the following:

  1. The family interview ruled out that poisoned people would consume alcohol before their last consumption - this information was added on line 91-92;
  2. In both the first and the second group, the age range is similar. The wide range of age depends on the real structure of drinking. 
    Extreme age values ​​are single, while the mean age for the studied groups
    is also equivalent. In contrast, detailed data added to the manuscript show that both groups are age comparable - age data is shown in lines 19, 22, and 86-88.
  3. The head space analysis protocol was precisely described with an indication of the possible identification of alcohol metabolites on lines 103-113.
  4. The correlation study between alcohol concentration and enzyme HEX and isoforms HEX A and HEX B in various types of fluids studied was made - table no. 2. Also statistical correlation between alcohol concentration and activity of enzymes in all studied materials was added – table no. 3 (line 170 and 172)

  5. In figure 1, the authors compared the enzymatic activity between both groups separately in all types of fluids, due to the Reviewer's suggestion (line 175).

    Yours sincerely

Round 2

Reviewer 2 Report

In the revised manuscript, authors have answered all the comments very well and incorporated appropriately in the manuscript.